# Evaluation of Interfering RNA Efficacy in Treating Hepatitis B: Is It Promising?

**DOI:** 10.3390/v16111710

**Published:** 2024-10-31

**Authors:** Giovana Paula Angelice, Pedro Henrique Roque, Gabriel Valente, Krishna Galvão, Livia Melo Villar, Vinicius Motta Mello, Francisco C. A. Mello, Bárbara Vieira Lago

**Affiliations:** Laboratory of Viral Hepatitis, Oswaldo Cruz Institute, Oswaldo Cruz Foundation, Rio de Janeiro 21040-900, Brazilliviafiocruz@gmail.com (L.M.V.);

**Keywords:** hepatitis B virus (HBV), chronic hepatitis, RNA interference (RNAi), gene therapy, covalently closed circular DNA (cccDNA)

## Abstract

Background: Despite an existing safe and effective vaccine for hepatitis B virus (HBV), it is still a major public health concern. Nowadays, several drugs are used to treat chronic hepatitis B; however, full healing remains controversial. The viral covalently closed circular DNA (cccDNA) formed by HBV forms a major challenge in its treatment, as does the ability of HBV to integrate itself into the host genome, which enables infection reactivation. Interfering RNA (RNAi) is a gene-silencing post-transcriptional mechanism which forms as a promising alternative to treat chronic hepatitis B. The aim of the present review is to assess the evolution of hepatitis B treatment approaches based on using RNA interference. Methods: Data published between 2016 and 2023 in scientific databases (PubMed, PMC, LILACS, and Bireme) were assessed. Results: In total, 76,949 articles were initially identified and quality-checked, and 226 eligible reports were analyzed in depth. The main genomic targets, delivery systems, and major HBV therapy innovations are discussed in this review. This review reinforces the therapeutic potential of RNAi and identifies the need for conducting further studies to fill the remaining gaps between bench and clinical practice.

## 1. Introduction

Hepatitis B virus (HBV) belongs to the virus family *Hepadnaviridae*, genus *Orthohepadnavirus* [1] and it is the major cause of liver diseases. HBV can be transmitted by bloodborne, sexual, and mother-to-child routes [2]. After a mean incubation period of 90 days, its clinical manifestations can range from asymptomatic to fulminant liver disease, depending on the host’s immune response [3,4]. Most adult patients with acute hepatitis B progress to convalescence, but approximately 5% of cases become chronic carriers [5]. Chronic infection is defined by HBV surface antigen (HBsAg) persistence for 6 months or more, with or without the detection of HBV antibody (anti-HBs). Chronicity risk increases by 30% to 50% when individuals are exposed to the virus up to the age of 5 years, and it can reach 90% when individuals are exposed to the virus in early childhood [6]. Chronically infected individuals are the main HBV reservoirs, and they face increased risk of liver cirrhosis and/or hepatocellular carcinoma (HCC) [5].

Despite the existing safe and effective vaccine against HBV, which has been available since 1981 [7], approximately 1.5 million new hepatitis B cases are reported on a yearly basis [2]. The World Health Organization (WHO) estimates that approximately 2 billion people have been infected with HBV worldwide and that 296 million people are chronic carriers. Although the vaccination rate has been increasing globally, hepatitis B was the cause of 820,000 deaths in 2019 [2,8]. Accordingly, HBV infection remains a major public health issue, and this scenario supports WHO’s efforts and global strategies to eliminate viral hepatitis by 2030.

WHO recommends the use of oral antiviral drugs, such as tenofovir (TDF) or entecavir (ETV), for the treatment of chronic hepatitis B, as they are the most potent means of suppressing HBV. While these drugs are highly effective in reducing and controlling viral load, disease relapse may occur upon interruption of antiviral therapy, as they do not promote viral clearance [2].

## 2. HBV Molecular Features and Replication

The HBV genome encompasses a circular partially double-stranded DNA molecule with ~3200 base pairs (bp), making it one of the smallest genomes among viruses infecting humans [9]. The longest negative DNA strand is complementary to viral RNAs. The shortest positive strand has a fixed 5′ terminus, while the 3′ terminus is variable, ranging from 50% to 90% the length of the template strand [9].

The virion, known as the “Dane particle”, consists of the genomic DNA coupled to an RNA-dependent DNA polymerase encapsidated by an icosahedral capsid, covered by a lipid envelope containing surface proteins (HBsAg) [10]. A large number of defective particles (~1013/mL in serum) composed of an empty envelope are released during HBV replicatio [11] n.

The virus adsorption occurs by HBV binding to heparan sulfate proteoglycan (PGHS) and sodium taurocholate (NTCP) receptors on hepatocytes. Following cell entry, the relaxed circular DNA genome (rcDNA) is transported to the cell nucleus, where it undergoes conversion into covalently closed circular DNA (cccDNA) mediated by a host-encoded DNA polymerase [8] as outlined in Figure 1.

The cccDNA is a highly stable molecule. This episome-like particle serves as the primary template for the transcription of pre-genomic and sub-genomic RNAs by the human enzyme RNA polymerase II. Despite the occurrence of viral integration into the host genome, most mRNA transcriptions depend on the cccDNA, which is the main viral molecular reservoir [12].

After structural protein synthesis and capsid assembly, pgRNA and HBV DNA polymerase are packed into the newly formed capsids, where pgRNA is reversely transcribed into partially double-strand DNA. Most nucleocapsids acquire envelope proteins via trans-Golgi and are released from the cell, while a small fraction is recycled to the nucleus to maintain high levels of cccDNA [13,14].

Furthermore, HBV presents surprisingly high-efficiency protein synthesis due to its unique genomic organization and partially overlapped open reading frames [9,15] as is outlined in Figure 2.

The unique characteristics of HBV, combined with its viral polymerase’s lack of proofreading ability and a notably high error rate (ranging from 1 × 10^5^ to 1 × 10^7^), contribute to its remarkable evolutionary pace, with a mutation rate of approximately 1.4 to 3.2 × 10^−5^ substitutions per site per year [16,17]. This rate is roughly 100 times faster than that of other DNA viruses, highlighting the virus’s significant capacity for genetic variation and adaptation [18]. This high variability reflects on the classification of ten genotypes (A–J) and several subgenotypes, based on divergences in the whole-genome sequence [19].

## 3. Current and Promising Treatments

The primary objective of hepatitis B treatment is to decrease the risk of progression of liver disease and its adverse outcomes, such as cirrhosis, hepatocellular carcinoma, and death. This treatment is targeted towards chronic cases, with the ideal goal of achieving sustained loss of HBsAg, leading to complete remission of inflammatory activity, although this outcome is rare. Alternatively, seroconversion to anti-HBe, reduction of viral load, and/or normalization of ALT are achievable outcomes in patients with persistent HBsAg and reactive or non-reactive HBeAg [4,20,21].

The established treatment comprises two main types of medications: cytokines with antiviral and immunomodulatory action, such as pegylated interferon alfa-2a (PEG-IFN-α), and nucleoside/nucleoside analogues (NAs), such as entecavir (ETV) and tenofovir disoproxil fumarate (TDF), each with different mechanisms of action and efficacy. While PEG-IFN-α acts by inhibiting viral replication and cell proliferation and modulating the immune response, NAs directly block viral replication. TDF is often chosen as the initial therapeutic option owing to its potent antiviral activity and high genetic barrier against resistance [21,22].

Despite therapeutic advancements, complete resolution of chronic hepatitis B infection and achieving a cure remain challenging due to the inability of current drugs to eliminate cccDNA from infected hepatocytes, requiring continued drug administration. While gene therapy studies have been conducted aiming to offer an alternative approach, the most effective strategy remains prevention through HBV vaccination, available worldwide, providing a high seroprotection rate of 90% to 95% in vaccinated adults and covering all virus genotypes [23].

The development of new therapeutic approaches for cccDNA inhibition is crucial, given its essential role in HBV clearance. Current gene therapy strategies include transcription activator-like effector nucleases (TALENs), zinc-finger nucleases (ZFNs), RNA-guided clustered regularly interspaced short palindromic repeats (CRISPR), and its variation by using the CRISPR-associated (Cas) enzyme [24].

Additional approaches for achieving complete viral clearance have been addressed. Small-molecule compounds known as capsid assembly modulators (CAMPs) aim to suppress HBV replication by disrupting proper capsid assembly, either by accelerating the formation of capsid-like particles or by inducing the formation of aggregated and aberrant capsid structures. These strategies are particularly attractive as capsids play a pivotal role in replenishing the cccDNA pool in cell nucleus and encapsulating pgRNA prior to reverse transcription [25,26].

Among the advanced drug candidates, MyrcludexB (Bulevirtide) is currently undergoing phase 3 trials. Acting as an N-terminal myristoyl truncated HBsAg, MyrcludexB represents the first entry inhibitor capable of competing with HBV for its receptor, thereby blocking infection in hepatocytes and contributing to viral transcriptional suppression [27,28].

## 4. Discovering an RNAi Pathway for Gene Silencing

RNA interference (RNAi) is a cellular gene-regulation mechanism whereby small RNA molecules post-transcriptionally inhibit gene expression by neutralizing targeted messenger RNA (mRNA) molecules [29].

Before the RNAi mechanism was fully understood and characterized as a gene-silencing process, it was initially referred to as “co-suppression” or “quelling”. These terms were used to describe early observations of post-transcriptional gene silencing, prior to the comprehensive elucidation of RNAi’s molecular underpinnings. In 1986, an inhibition process mediated by antisense RNA was first observed in plant cells, demonstrating their ability to suppress the expression of target genes [30,31].

Subsequently, in 1991, the effective and specific suppression of target genes by antisense RNA was demonstrated in a study involving nematodes [32]. The term “RNA interference” was coined in 1998 following the discovery of double-stranded RNA as a potent gene-silencing agent. Unlike endogenous mRNA injection, the double-stranded RNA was found to be highly effective in regulating protein production in *Caenorhabditis elegans* [33]. Two years later, the RNAi mechanism was reported in mammals for the first time, successfully silencing three maternally expressed genes in mice [34].

RNAi was first proposed as a potential treatment for chronic HBV infection in 2008 [35,36]. Recently, studies carried out with ARC 520, an RNAi-based therapeutic designed to treat HBV, have demonstrated promising outcomes, notably in the reduction of viral RNA and protein levels. This drug targets cccDNA derived from viral replication, exhibiting high specificity. By silencing proteins crucial to the HBV replication cycle, ARC-520 effectively interrupts viral replication [37].

Small interfering RNA (siRNA) is a pivotal molecule in RNAi technology, playing a fundamental role in gene silencing. It consists of small double-stranded RNA sequences, typically 20 to 25 nucleotides in length, which bind to complementary mRNA sequences, promoting the degradation of the target mRNA and preventing the translation of proteins involved in viral replication or disease development [38].

The RNAi process can be divided into three main steps. Initially, long double-stranded RNA (dsRNA), either introduced or expressed within the cell through base pairing of sense and antisense transcripts or the formation of stem-loop structures, is processed into small RNA duplexes by the enzyme Dicer, a type of ribonuclease III [39]. These duplexes are then unwound, and one of the strands is preferentially incorporated into the RNA-induced silencing complex (RISC). This loaded single-stranded RNA, known as the guide strand, enables the RISC to scan the transcriptome and identify potential target RNAs. The guide strand directs the RISC’s endonuclease, an Argonaute protein, to cleave mRNAs containing sequences homologous to the guide strand, repeating the process over multiple cycles [39,40,41] This precision enables RNAi to exert post-transcriptional gene silencing, halting protein production from the target mRNA [29].

Despite these advantages, RNAi faces certain limitations. Mutations in the viral genome or genotypic variations can diminish the efficacy of siRNA, leading to the emergence of escape mutants. Moreover, siRNA is inherently unstable in the extracellular environment and prone to degradation by nucleases. Nevertheless, advancements in siRNA chemical modification and delivery methods have broadened its application, particularly in therapeutic contexts such as hepatitis B treatment.

## 5. RNAi Pathway

While numerous classes of small RNAs have been described, there are three main categories to be assessed: short interfering RNAs (siRNAs), microRNAs (miRNAs), and piwi-interacting RNAs (piRNAs). This review concentrates on miRNAs and siRNAs, highlighting their functions as post-transcriptional regulators and gene transcription inhibitors, respectively [42,43,44].

siRNA and miRNA have distinct cell origins and exhibit differences in their double-stranded RNA (dsRNA) precursors. siRNAs can originate from viruses, transposons, and transgenes, and their duplexes show perfect base pairing. In contrast, miRNAs are derived from the eukaryote genome. There are more than a thousand miRNAs in the human genome regulating at least 30% of genes involved in cell growth control, tissue differentiation, heterochromatin formation, and cell proliferation [42,43,44].

Despite differences in origins and functions, miRNAs and siRNAs share a common action mode: The minimal effector is a ribonucleoprotein complex, comprising an Argonaute family protein bound to a small RNA through base-pairing interactions with the target gene. This complex, the RISC, facilitates target mRNA silencing via transcriptional degradation and/or repression [42,43,44].

The application of RNAi as a treatment strategy for chronic hepatitis B has gained prominence due to the unique genomic and molecular characteristics of HBV. Figure 3 illustrates how this strategy works.

## 6. Materials and Methods

The literature search was performed using the following databases: PubMed, PMC, LILACS, and BIREME. PubMed and PMC are open-access bibliographic repositories developed and maintained by the National Center for Biotechnology Information (NCBI), which comprise life science and biomedical journals. LILACS is an index and bibliographic repository of scientific and technical production in health sciences published in Latin America and the Caribbean. BIREME is a specialized center holding a bibliographic collection focused on medicine and health sciences; this center is part of the Pan American Health Organization/World Health Organization (PAHO/WHO).

The search was conducted separately on each platform. The following descriptors were taken into consideration for the database search: “small interfering RNA”, “shRNA”, “siRNA”, “RNAi”, “RNA interference”, “genomic silencing”, “hepatitis”, “HBV”, and “hepatitis B virus”. Logical operators “AND” and “OR” were employed to combine descriptors to the aforementioned terms and refine the search criteria to focus on articles addressing hepatitis B virus silencing through RNA interference. The search was limited to articles published between 2016 and 2023.

The initial search retrieved 76,949 articles (PubMed = 577, PMC = 75,404, LILACS = 163, and BIREME = 805), and following a preliminary screening of titles, the majority of the articles were excluded due to irrelevance to the topic. This significantly reduced the number of selected papers. Subsequently, the remaining articles were peer-reviewed, and their titles and abstracts were read thoroughly to confirm relevance to the proposed research. This process ultimately resulted in 226 articles being selected for detailed review and inclusion (Table 1).

In the third step, all articles were read in full, resulting in 25 articles meeting the study’s inclusion criteria. Duplicated articles, articles unrelated to the proposed topics, review articles, or articles that were not freely accessible in digital media were excluded. Additionally, several aspects were considered during the analysis of the selected articles, including country of publication, genomic region affected by silencing, virus genotype, RNAi delivery system, target organ, study phase, cell or animal model, number of individuals, and silencing.

## 7. Overview

### 7.1. Genes and Genotypes

The strategy of targeting overlapping genomic regions allows the silencing of two or more ORFs with a single interfering molecule due to the structure of the HBV genome. Additionally, this strategy can suppress the reverse transcription of pgRNA in genomic DNA. Nevertheless, designing siRNAs and microRNAs against conserved regions is essential to ensure both pangenic and pan-genotypic activity, as gene silencers can become ineffective due to single point mutations or intergenotype variability.

Most studies targeted the X gene of the HBV genome, which encodes the multifunctional protein X. This protein is involved in viral replication and proliferation processes [45], in inflammation and immunomodulation [46,47], and in regulating the transcription of genes involved in hepatocarcinogenesis [48,49,50]. Although protein X itself is not essential for viral viability, the three viral mRNAs and the pre-genomic RNA overlap in this region [45,51]. This makes the X gene an attractive target, as a single siRNA can potentially silence all viral genes simultaneously.

In total, 15/25 (60%) articles adopted this strategy (Table 2). A late-stage study (ARB-1740) used different shRNAs, with different targets in HBV genome (POL/X and Pre-Core/X) overlapping regions.

Regarding genetic variability, the main HBV silencing drugs in trials (ARC-520 and ARB-1740) were pan-genotypic. Most studies among those that have analyzed specific HBV genotypes included HBV/A, HBV/B, HBV/C, or HBV/D, which have significant impact on the pathophysiology of the disease. Genotypes A and D are the most prevalent worldwide [4]. In addition, HBV/A is associated with a direct progression to liver cancer, without the usual intermediate cirrhosis stage [75]. Genotypes B and C prevail in Asia and Oceania [76]; however, genotype C in Asian populations is related to increased risk of hepatocellular carcinoma in comparison to genotype B [75].

In addition to assessing HBV silencing effectiveness, one of the studies validated the functionality of ARB-1740 against hepatitis Delta virus (HDV) [54]. HDV, a defective virus, requires HBV for replication, utilizing HBsAg to enter hepatocytes and to assemble new particles. Co-infection HBV/HDV is considered the most severe form of hepatitis, with no cure or effective treatment available. RNAi therapy can be a promising treatment strategy, as HBV silencing consequently affects the HDV replication cycle [76].

### 7.2. Delivery Systems

Lipid nanoparticles (LNPs), adenoviral vectors, adeno-associated viruses (AAVs), and lentiviral vectors (LVs) were the main in vivo delivery systems described in the selected articles. LNP is the most often employed method among the herein assessed ones (e.g., ARB-1740) because it allows intracellular uptake, optimal siRNA targeting to hepatocytes, and protection against plasma degradation [53,77,78,79]. However, nanoparticles face several challenges, including avoiding aggregation with blood and extracellular components, rapid elimination via renal filtration and urinary excretion, and the risk of activating an innate immune response [80,81,82].

Adenoviral vectors and AAVs have demonstrated efficacy in delivering and expressing siRNAs. However, implications with systemic immunostimulation and the need for sustained expression remain limiting factors. In this context, LVs might offer a potential solution, as they can integrate into the host cell genome, promoting sustained and potent gene expression. Furthermore, LVs can mediate stable transduction in cells at different mitotic states, which allows for targeting cells that have low or no division after cell differentiation [63,79,83]. However, in vivo tests showed low efficiency of LVs in transducing adult hepatocytes, and their genomic integration poses an oncogenic risk requiring lifelong monitoring [81].

The Multifunctional Envelope-type Nano Device (MEND) is among the alternative delivery methods. It is a non-viral vector comprising RNAi molecule nanocarriers, a lipid envelope, and a plasmid DNA condensed by poly-L-lysine (PLL). The MEND with siRNAmix (MEND/HBV-siRNAmix) has shown high specificity for hepatocyte delivery and greater efficacy than ETV in controlling HBV, as it reduces HBsAg, HBeAg, and HBV DNA levels both in vitro and in vivo. A single dose of the MEND/HBV-siRNA mix promoted the inhibition effects on HBV for approximately 14 days [58]. Therefore, the MEND’s transfection activity is comparable to that of adenovirus, but it may present greater cell–transfection heterogeneity. In addition, the MEND has no toxicity [59].

The recombinant preS1 protein with truncated protamine (tP) (preS1-tP proteins) is another promising delivery method. The preS1 protein is crucial for cell entry during natural HBV infection, and its entry efficiency is not limited by the effects of neutralizing anti-HBs antibodies. Zeng et al. (2018) [74] evaluated preS1-tP proteins targeting HBV, demonstrating high suppression of HBV mRNAs, HBsAg, HBeAg, and HBV DNA, as well as inhibition of cccDNA in HepG2.2.15-NTCP cells. Using preS1 proteins or recombinant preS1-tP in nanoparticle delivery can be promising in HBV depuration mainly due to their inhibitory action on HBV cccDNA [51,74].

Ivacik et al. (2015) proposed the production of LVs with liver-specific RNA polymerase II cassettes to produce HBV-silencing artificial primary miRNAs (pri-miRNAs). These amiRNAs are derived from, or mimic, pri-miR-31, which is an anti-HBV microRNA. They are generated in monocistronic or tricistronic cassettes designed to silence one or three sites in the conserved region of the HBV X gene [63]. Additionally, expression cassettes combining RNAi and miRNAs, such as the pHsa-miR16-siRNA vector expressing human miR-16 and HBV X siRNA, have shown significant inhibitory effects against HBV compared to single-expression vectors [62].

Finally, Wang et al. (2017) used the gRNA-miRNA-gRNA ternary cassette in combination with CRISPR/Cas9 to achieve RNAi-mediated HBV inhibition. This approach resulted in efficient suppression of genotypes A to D; reduction of HBsAg, HBeAg, and HBcAg levels; and elimination of cccDNA [84].

## 8. Innovative Works

RNAi presents high therapeutic potential for inhibiting HBV and integrated HBV DNA transcripts. The main advantages of RNAi include high specificity, safety, and efficacy in gene silencing, as demonstrated in previous studies [85].

ARC-520 was the first RNAi-based therapy against HBV to reach clinical trials. It consists of two synthetic interfering RNAs targeting the ORF X. In this therapy, RNAi molecules are conjugated to cholesterol to enhance delivery efficiency to hepatocytes [37,57,86]. ARC-520 aims to reduce cccDNA-derived transcripts, thereby silencing viral antigens and decreasing HBV DNA levels. The inclusion of two siRNAs is intended to reduce the risk of viral escape and to provide broader coverage across different genotypes and viral variants [37,57].

Several studies have evaluated the safety, tolerability, dosing, and pharmacokinetics of ARC-520 in humans. Serological, virological (in serum and liver), and histological profiles of patients with chronic HBV infection after single and multiple doses were also assessed [37,55,56,57,75,87]. Results indicated significant suppression of viral antigens, such as HBsAg and HBeAg, as well as HBV DNA. The HBsAg suppression persisted for 4 to 12 weeks after a single or multiple doses, respectively [75]. Although ARC-520 remains at the clinical testing phase, it is considered a highly promising tool. It has the potential to restore effective immunity in hosts and increase functional cure rates through sustained HBsAg loss, with or without seroconversion [37,57,86].

ARB-1740 is another RNAi-based antiviral agent currently in clinical trials. This agent comprises three siRNAs delivered to hepatocytes using lipid nanoparticle technology. These siRNAs target three highly conserved regions of the HBV genome, inhibiting the selection of escape mutants and increasing efficacy against HBV [53]. Studies have demonstrated that ARB-1740 was successful in silencing viral elements, such as HBsAg, HBeAg, HBcAg, and the HBV genome [53]. ARB-1740 was tested in vitro and in vivo for HBV and, for the first time, for HDV inhibition. Although HDV animal models are not well established, tests conducted on three mice showed satisfactory HDV inhibition rates; nonetheless, further studies are still needed to assess the applicability of this technology focused on HDV [53,54].

Recent pre-clinical studies have shown that the combination of the siRNA JNJ-3989 with nucleotide analogues (NA) significantly reduces levels of HBsAg, HBV RNA, HBeAg, and HBcrAg. This treatment, involving three doses of JNJ-3989, promoted sustained reduction of all viral markers in a subgroup of patients up to 336 days after the last dose. While this combination demonstrated a good safety profile and persistent HBV inhibition in many patients, further studies are essential to fully understand its potential [60,61].

Advancements in new viral agents, such as RNA interference combined with N-acetylgalactosamine (GalNAc), have shown promising outcomes in both pre-clinical and clinical phases. VIR-2218 is a GalNAc-conjugated siRNA designed to specifically target nucleotides 1577 to 1596 in the HBV genome, since most HBV ORFs are overlapped at this position. This siRNA employs Enhanced Stabilization Chemistry (ESC) technology, which consists of 2′-deoxy-2′-fluoro, 2′-O-methyl ribose sugar modifications, and phosphorothioate (PS) backbone modifications. These modifications allow higher metabolic stability and accumulation in acidic intracellular compartments. This process results in a prolonged pharmacodynamic effect. VIR-2218 is currently in the pre-clinical phase and has been associated with dose-dependent reductions in HBsAg levels [67,68].

Alternative RNAi approaches have also been explored with notable success using multimeric short RNA (shRNA) constructs derived from pri-miR-31. These pri-miRs, in conjunction with the liver-specific modified murine transthyretin promoter (mTTR), targeted the HBx region and were delivered using recombinant adeno-associated viruses (scAAVs). This strategy resulted in viral marker suppression for at least 32 weeks without evidence of innate toxicity or immunostimulation [83]. Similarly, Ivacik et al. (2015) demonstrated the efficacy of pri-miRs using a lentiviral delivery system, achieving sustained HBV replication inhibition without any evidence of hepatotoxicity [63].

A novel approach combining miRNA and siRNA technologies utilized a dual-expression vector, pHsa-miR16-siRNA, to express both human miR-16 and HBV X siRNA. This combination significantly inhibited viral replication and reduced HBV mRNAs and genomic DNA in the HepG2.2.15 cell line [62].

An innovative study demonstrated the potential of synthesizing therapeutic RNAi using miRNA biosynthesis machinery in lettuce. SiRNA 471 and siRNA 519 were expressed in lettuce plants, which are easy to grow and produce substantial biomass. Mice fed with a lettuce decoction at a lower concentration containing these synthetic edible miRNAs showed similar inhibitory effects on hepatocytes as those observed with synthetic RNAi at normal levels, without toxicity over a 15-month treatment period [64].

RNAi technology has also been associated with other technologies to overcome barriers and improve its efficiency. The shRNA expression combination in mesenchymal stem cells led to therapeutic efficiency in treating liver injury and in suppressing HBV expression in tissue [71].

An intriguing approach combining CRISPR/Cas9 and RNAi utilized a ternary cassette expressing two guide RNAs against HBV and one mimic of the pre-miR-31 anti-HBV. This strategy effectively inhibited HBV replication in genotypes A to D in vitro and in vivo and further disrupted HBV genomic DNA and cccDNA in vitro [84]. Recent findings on HBV silencing progress through RNAi are summarized in Table 2.

## 9. Available RNAi-Based Drugs: Current Overview

Currently, the FDA has approved four siRNA medications: Patisiran, Givosiran, Lumasiran, and Inclisiran. Notably, Givosiran is distinguished by its liver-targeted therapy, specifically designed to the treatment of acute hepatic porphyria (AHP). AHP is a rare hereditary disorder that originates in the liver due to a disruption in the heme synthesis pathway, leading to the accumulation of toxic porphyrin precursors, which cause severe abdominal pain, neurological manifestations, and other systemic complications [88].

Despite these advancements, siRNA therapies for viral infections have yet to receive FDA approval. However, several siRNA- and miRNA-based therapeutics are in advanced clinical trials targeting viral infections, with a particular focus on treating chronic viral infections. These approaches are being investigated for conditions such as hepatitis B, human papillomavirus (HPV)-related cancers, and hepatitis C [88].

The four FDA-approved RNAi-based drugs have demonstrated efficacy in treating their specific indications but are also associated with significant adverse effects. Commonly reported side effects include injection site reactions, nausea, fatigue, and elevated transaminases levels, with instances of anaphylactic reactions. In particular, Onpattro has been linked with infusion-related reactions such as flushing, back pain, and dyspnea, especially during initial infusions [88,89]. These adverse effects underscore the immune response elicited by the body and highlight the challenges inherent in RNAi therapy, including the optimization of delivery systems and the reduction of off-target effects [89].

In comparison, RNAi-based drugs under development for the treatment of hepatitis B, such as ARC-520, VIR-2218, and ARB-1740, have exhibited safety profiles similar to those of approved drugs. Preliminary studies reveal that these agents encounter similar challenges, including injection site reactions, elevated transaminases and bilirubin, and systemic effects such as headache, upper respiratory infection (URI), drowsiness, lethargy, elevated creatinine levels, nausea, and fatigue [37,53,57,68].

## 10. Limitations and Future Problems

The therapeutic potential of RNAi has been assessed to better understand its applications and limitations. Although research on HBV treatment using RNAi is gradually advancing, some obstacles still must be overcome. The clinical application of RNAi therapies is primarily constrained by issues of stability, delivery efficiency, and potential toxicity [64].

Many studies included in this review have evaluated promising solutions for the development of RNAi-based therapies. Utilizing multiple synthetic RNAs to inhibit HBV expression has shown effectiveness in achieving pan-genotypic silencing and preventing escape mutations. However, it is essential to thoroughly assess toxicity, ensure the inhibition of all viral transcripts, and address off-target effects that may occur during treatment [56,57,60,84,85,86,87].

Despite being at the clinical stage, ARC-520 does not promote full silencing of HBsAg transcripts from integrated HBV DNA, indicating the need for further refinement to suppress all viral products effectively [57]. Furthermore, the delivery of RNAi triggers and the maintenance of their expression over prolonged periods, ensuring a sustained therapeutic response, are the main current challenges. Due to their small size, hydrophilicity, and negative charge, RNAi triggers require transporters for delivery, as they cannot easily interact with cell membranes and are rapidly cleared from the body.

Delivery systems employing expression vectors based on lentiviruses, adeno-associated viruses, lipid conjugation, or lipid nanoparticles represent promising approaches for achieving long-term expression in mammals. However, safety concerns, particularly regarding the potential toxicity of these delivery molecules, must be carefully considered [85,90].

Finally, HBV genetic variability must be considered when validating new RNAi-based treatments. HBV genotypes can impact the host responses to treatment. Studies have shown that HBV genotypes influence the efficacy of interferon therapy, with genotypes D and C exhibiting a weaker response to interferon alpha and pegylated interferon alpha-2b therapy compared to genotypes A and B [75,91]. This difference was also observed in lamivudine treatment, where genotype D carriers demonstrate a better response than those with genotype A [92]. Moreover, in tenofovir-treated patients with a natural history of HBV immune clearance, HBsAg loss was almost exclusively limited to Caucasian patients infected with HBV genotypes A and D and was rarely observed in Asian patients infected with genotypes B and F [93].

In this context, although RNAi-based treatments introduce a novel mechanism compared to traditional therapies, the genetic diversity of HBV remains a critical factor influencing treatment efficacy. This genotypic variability represents a potential limitation in the approval and widespread application of new RNAi-based drugs. Additionally, mutations occurring in RNAi sequences or within the integrated HBV genome can significantly diminish or completely negate the effectiveness of these therapies. To mitigate this issue, most RNAi therapies employ small interfering RNA (siRNA) sequences designed to target highly conserved regions of the viral genome, aiming to ensure cross-genotype compatibility and achieve a pan-genotypic effect, thereby minimizing the risk of escape mutations.

## 11. Conclusions

RNA interference (RNAi) has emerged as a potentially transformative therapeutic strategy for the management of hepatitis B, despite not constituting a definitive cure. This approach holds promise by addressing the shortcomings of existing treatments, which fail to effectively target the covalently closed circular DNA (cccDNA) and integrated HBV sequences within the host genome. RNAi’s ability to specifically inhibit the early phases of HBV replication and induce post-transcriptional gene silencing confers substantial therapeutic benefits. Moreover, the presence of overlapping open reading frames (ORFs) within the HBV genome allows RNAi to concurrently suppress the expression of multiple viral proteins, thereby enhancing its therapeutic potential. Several RNAi-based drugs against hepatitis B are currently in advanced stages of clinical testing. These drugs exhibit pan-genotypic potential, offering a promising strategy for the treatment of HBV infection. However, despite the demonstrated advantages, several critical aspects require further investigation. These include the optimization of delivery vectors, the development of strategies to mitigate the emergence of point mutations in target regions, the assurance of sustained expression of shRNAs, and the minimization of off-target effects to enhance treatment specificity.

Furthermore, siRNA inhibitory effects can be complemented and enhanced by other compounds, which not only suppress viral expression but also reduce liver damage and restore the host’s immune response. While this review reinforces the significant therapeutic potential of RNAi, further studies are warranted to address current limitations and explore the potential of this mechanism in achieving higher rates of functional cure of chronic hepatitis B.

## Figures and Tables

**Figure 1 viruses-16-01710-f001:**
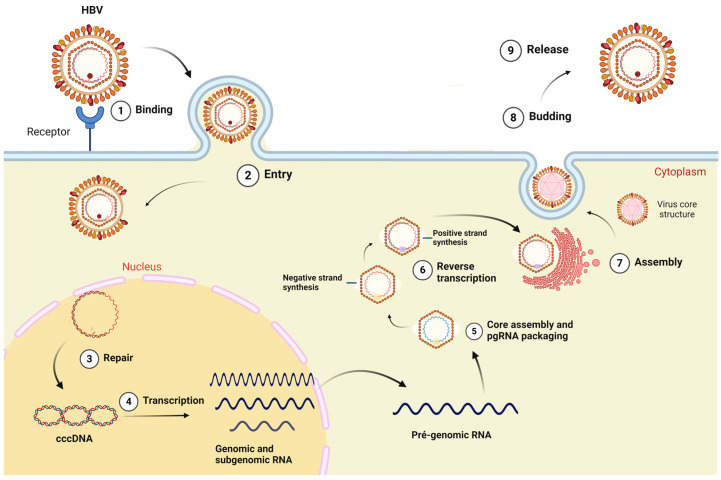
HBV replication cycle model. The viral particle enters the cytoplasm by binding to the sodium taurocholate receptor cotransporting polypeptide (hepatocyte membrane receptor); the capsid is transported to the nucleus, wherein the HBV DNA is released. The HBV genome is converted to covalently closed DNA (cccDNA) which is used sequentially as transcriptional template for genomic and sub-genomic RNAs. Transcripts are exported to cytosol, and they can be used as messenger RNAs (mRNAs). Pre-genomic RNA (pgRNA) works as replication template. The new viral particle is formed at the transition between the endoplasmatic reticulum and the Golgi complex, and it is then secreted. Created at Biorender.com (accessed on 2 August 2024).

**Figure 2 viruses-16-01710-f002:**
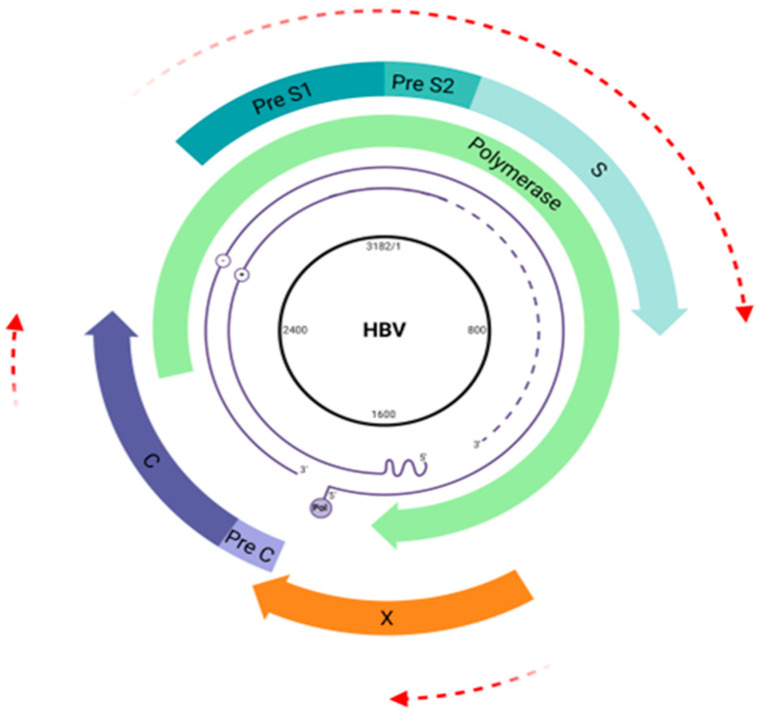
Hepatitis B virus (HBV) genome map. The HBV genome is a double-stranded DNA (3.2 kb) comprising four overlapping open reading frames (ORFs) coding the viral envelope (pre-S1/pre-S2/S) (blue arrow), core proteins (pre-C/C) (purple arrow), viral polymerase (green arrow), and HBx protein (orange arrow). Overlapping regions are highlighted by red arrows. Created at Biorender.com (accessed on 2 August 2024).

**Figure 3 viruses-16-01710-f003:**
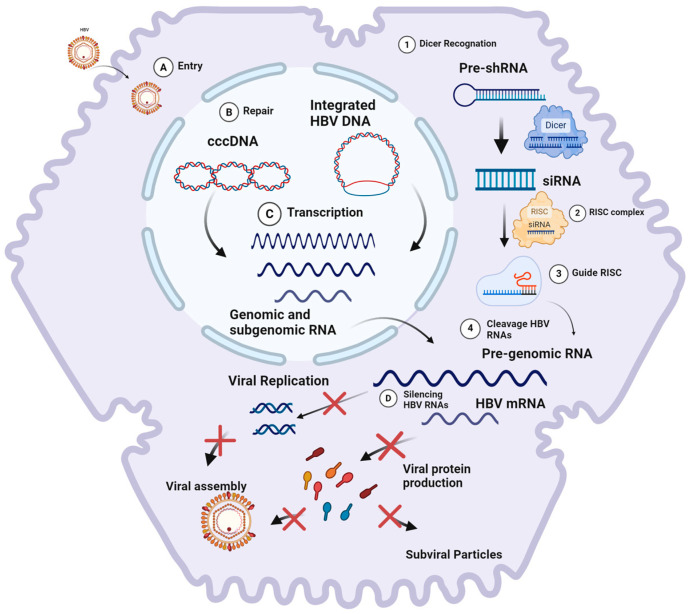
(1) The DICER enzyme recognizes and processes exogenous HBV mRNA (pre-shRNA) into small RNA fragments upon initial contact with a eukaryotic cell. (2) These small RNA fragments are subsequently incorporated into the RNA-induced silencing complex (RISC) as guide strands. (3) Upon subsequent interaction with exogenous mRNA, the guide strand within the RISC recognizes the complementary sequence. This recognition induces conformational changes in the RISC, leading to the cleavage of the target mRNA. Cellular endonucleases then degrade the mRNA residues, thereby preventing the synthesis of encoded proteins. (4) The RISC, utilizing the interfering RNA, cleaves HBV pre-genomic RNA and mRNA, thereby silencing viral replication and protein production. The HBV replication process involves the following steps: (A) viral entry into hepatocytes via endocytosis; (B) release and transport of the nucleocapsid to the nucleus, where the HBV genome is converted into covalently closed circular DNA (cccDNA); (C) transcription of cccDNA, resulting in the production of pre-genomic RNA and mRNAs necessary for viral protein synthesis and replication; and (D) RNA interference (RNAi) that targets pre-genomic RNA (pgRNA) and mRNAs, thereby halting viral protein synthesis and replication. Created at Biorender.com (accessed on 2 August 2024).

**Table 1 viruses-16-01710-t001:** Screening of articles selected for the review.

Stages	Investigator 1	Investigator 2	Investigator 3	Investigator 4	Total
First	805	577	75,404	163	76,949
Second	58	81	49	38	226
Third	42	39	45	36	162
Fourth	6	6	10	3	25

**Table 2 viruses-16-01710-t002:** Recent findings on HBV silencing progress caused by RNAi.

Candidates	Target	Genotypes	Delivery	Research Phase	Silencing Efficacy	Others
3p-siHBx [52]	Gene X	NI	RNA complexed with 1,2-dioleoyl-3-trimethylammonium-propane	Pre-clinical	HBV replication inhibition in HBV-carrying mice; dendritic cell maturation induction, natural killer cell activation, and reverse HBV-induced CD8+ T cell exhaustion	-
ARB-1740 [53,54]	Genes Pol/X and Pre-Core/X	Pan-genotypic	LNP	Pre-clinical	Powerful viral proteins control in blood and liver in different HBV models; HBsAg viral envelope protein reduction; high innate immune response	One of the studies validated the ARB-1740 functioning in hepatitis Delta virus (HDV) silencing by showing both HDV reduction and HBV replication
ARC-520 [37,55,56,57]	Genes S and X	Pan-genotypic	RNAi + NAG-MLP	Clinical	Moderate HBsAg inhibition	In association with Entecavir, ARC-520 reduced HBsAg, HBeAg, HBc, and HBV RNA by undergoing two HBeAg seroconversions. Tests combined to Entecavir.
MEND/HBV-siRNA [58,59]	Genes Pre S/S, Pre/Core, and Pol	A, B, and C	MEND—pH-sensitive multifunctional envelope-type nanodevice	Pre-clinical	Efficient HBsAg and HBeAg reduction in vitro and in vivo	MEND/HBV-siRNA can control HBV more efficiently than ETV, as shown in one of the studies. It can promote efficient siRNA delivery and liver-specific delivery.
JNJ-3989 [60,61]	All transcripts	NI	Subcutaneous injection—RNAi associated with GalNAc	Clinical	Significant HBsAg, HBV RNA, HBeAg, and HBcrAg reductions	Tests combined to the nucleos(t)ide analogue, with/without capsid assembly modulator JNJ-56136379.
pHsa-miR16-siRNA [62]	Gene X	A, B, and C	pHsa-miR16-siRNA vector	Pre-clinical	Inhibition of HBV expression genes; inhibitory HBsAg and HBeAg rates; HBV mRNA decrease, genomic HBV DNA, and the copy number of the HBV DNA supernatant	Tests combined to two RNAi methods (siR-1583 + pmiR-16).
pri-miR-31 [63]	Gene X	A-D	Lentivirus	Pre-clinical	Effect on inhibiting HBV replication and on destroying HBV genome in vitro and in vivo	Tests combined to CRISPR.
siR471 and siR519 [64]	Gene S	NI	Plant-derived artificial microRNAs	Pre-clinical	HBsAg expression inhibition	-
siRNA-X, siRNA-P and siRNA-C [65]	Genes X, P, and C	NI	LNP with Pre/S1 peptide	Pre-clinical	HBV DNA reduction, mainly in the P and X gene siRNAs	-
SR16-X2M2 [66]	Genes X and S	NI	LNP	Pre-clinical	-	-
VIR-2218 [67,68]	All transcripts	Pan-genotypic	RNAi conjugated with GalNAc (N-acetylgalactosamine)	Clinical	HBsAg reduction	Enhanced Stabilization Chemistry Plus (ESC+)—chemically modified RNAi.
shRNA-HBV [69]	Gene X	Pan-genotypic	AAV	Pre-clinical	The best results were recorded for the AAV8 vector, which co-expresses anti-HBV shRNA and TuD and generates a more robust and persistent HBV knock-down	-
msiHBx [70]	Genes S and X	C	Uridine-bulge siRNA	Pre-clinical	HBsAg and HBV DNA decrease	-
Lx-shRNA157-1694 [71]	Genes S and X	NI	AAV	Pre-clinical	Suppression of HBV expression in the liver	-
siRNA-loaded YSK13-C3-LNPs [72]	-	A, B, and C	LNP with pH-sensitive cationic lipids	Pre-clinical	Reduction in both HBV DNA and its antigens, without toxicity	-
HBV-shRNA [73]	Genes S, X, and C	NI	Chimeric adenovirus–HBV vectors		Caused significant reduction in viral antigens, RNAs, and circulating enveloped virus particles	HBV shRNA vectors mainly proved effective in an infection model, in vivo.
HBV NLS siRNAs [74]	-	NI	Fusion proteins containing one single chain variable fragment (scFv) and recombinant preS1-truncated protamine(tP)	Pre-clinical	Inhibition of HBV’s mRNA expression; HBsAg and cccDNA level suppresion	Recombinant preS1-tP proteins successfully carried the HBV NLS siRNAs to the HepG2.2.15 cells.

NI: Not informed.

## Data Availability

All relevant data are present in the manuscript.

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
