# Peer review of "Evaluation of Interfering RNA Efficacy in Treating Hepatitis B: Is It Promising?"

_viruses, 2024, doi:10.3390/v16111710_

Round 1

Reviewer 1 Report

Comments and Suggestions for Authors

This is a fine review regarding the uses of RNA silencing technologies to control HBV infection. It would be interesting to note what side effects are taking place within clinical trials, when we may see RNAi based drugs released for the general public, etc. 

Are there RNAi based drugs available for virus infections right now? Could you list a few of them? That would provide the reader with a sense of how far this technology has come. 

Thank you for writing this interesting review!

Author Response

Dear Editor,

We hereby submit the revised version of our manuscript “Evaluation of Interfering RNA efficacy in Treating Hepatitis B: is it Promising?”.

Thank you your valuable comments and suggestions. We believe that the manuscript has gained significantly from all of the modifications. Our detailed point-by-point responses to the reviewers' comments are below.

If you need further information, please do not hesitate to contact me.

Thank you for taking the time to review our work.

Sincerely,

Pedro Henrique Roque da Conceição

Comments 1: This is a fine review regarding the uses of RNA silencing technologies to control HBV infection. It would be interesting to note what side effects are taking place within clinical trials, when we may see RNAi based drugs released for the general public, etc.

Response 1: Thank you for your valuable feedback on our review. We have added a new paragraph (line 426 ,Topic: Available RNAi-Based Drugs: Current Overview) discussing the side effects observed in clinica ltrials and the anticipated timeline for RNAi-based drugs to become available to the general public.

Comments 2: Are there RNAi based drugs available for virus infections right now? Could you list a few of them? That would provide the reader with a sense of how far this technology has come.

Response 2: Currently, the U.S. Food and Drug Administration (FDA) hasapproved four RNAi-baseddrugs, one of which targets liver disease (Gavin M. Traberand Ai-Ming Yu, 2022). However, none of these are specifically for viral infections. Nonetheless, several siRNA and miRNA-based therapeutics are in clinical trials with the goal of treating viral infections. Beyond hepatitis B, these approaches are also being explored for human papillomavirus (HPV)-related cancers and hepatitis C.

Reviewer 2 Report

Comments and Suggestions for Authors

The hepatitis B virus (HBV) is a major public health problem. The drugs available to treat chronic hepatitis B cannot eradicate the infection because the viral genome remains in the hepatocytes as covalently closed circular DNA (cccDNA) or is integrated into the host genome.

Based on a comprehensive literature search, the authors describe the most promising interfering RNA (RNAi) molecules specifically targeting the HBV genome that have been produced in the last decade. The article describes in detail the main genomic targets of RNAis and their therapeutic potential. The authors also recognize the limitations of RNAis themselves and their delivery methods, which still face formidable challenges before their use in clinical practice.

The article is well-written and offers no major criticisms. However, it is recommended to better emphasize that even RNAi, like conventional drugs, cannot eliminate the infection. RNAi are also very sensitive to viral mutations, which are very common, and, depending on the sequence conservation of the target sequences, may not work equally well with all HBV genotypes. These three major limitations should be discussed in sections 9 and 10.

Author Response

Dear Editor,

We hereby submit the revised version of our manuscript “Evaluation of Interfering RNA efficacy in Treating Hepatitis B: is it Promising?”.

Thank you your valuable comments and suggestions. We believe that the manuscript has gained significantly from all of the modifications. Our detailed point-by-point responses to the reviewers' comments are below.

If you need further information, please do not hesitate to contact me.

Thank you for taking the time to review our work.

Sincerely,

Pedro Henrique Roque da Conceição

Comments 1: The hepatitis B virus (HBV) is a major public health problem. The drugs available to treat chronic hepatitis B cannot eradicate the infection because the viral genome remains in the hepatocytes as covalently closed circular DNA (cccDNA) or is integrated into the host genome.

Based on a comprehensive literature search, the authors describe the most promising interfering RNA (RNAi) molecules specifically targeting the HBV genome that have been produced in the last decade. The article describes in detail the main genomic targets of RNAis and their therapeutic potential. The authors also recognize the limitations of RNAis themselves and their delivery methods, which still face formidable challenges before their use in clinical practice.

The article is well-written and offers no major criticisms. However, it is recommended to better emphasize that even RNAi, like conventional drugs, cannot eliminate the infection. RNAi are also very sensitive to viral mutations, which are very common, and, depending on the sequence conservation of the target sequences, may not work equally well with all HBV genotypes. These three major limitations should be discussed in sections 9 and 10.

Response 1: Thank you for your valuable feedback ono ur review. We have added new paragraphs (topic 9, line 421 -447, and topic 10, lines 481 - 490) in which we discussed the limitations related to the genetic variability of HBV (emergence of point mutations and genotypic variability) in the efficacy of RNAi-based therapies, reinforcing the importance of  target highly conserved regions in HBV genome.

In addition, we clarify that RNAi, like conventional drugs, cannot provide viral clearence.

Reviewer 3 Report

Comments and Suggestions for Authors

Dear authors,

Interesting topic which I enjoyed reading.

However, my main concern is that although the focus of your Review is on RNAi and its potential use in the treatment Hep B, the description of RNAi is probably the lowest standard within the manuscript's text.

Many incorrect statements have been made, or a lack of detail has been provided, and together, to me this demonstrates a lack of understanding of small RNA biology and RNAi field by the authors.

There is also numerous other English language issues which require addressing / correction in a revised manuscript version.

Figure 3 can be removed as it is inaccurate and fails to provide any additional insight into the general field.

Figure 4 needs work / redoing in order to be mechanistically accurate in the information it is attempting to present.

Please use the annotated copy of your article as a template for correction/amendment as part of producing a revised manuscript version.

Comments on the Quality of English Language

English is quite good overall.

But a thorough revision of word choice / sentence structure is required throughout all sections.

Author Response

Comments 1: 

Dear authors,

Interesting topic which I enjoyed reading.

However, my main concern is that although the focus of your Review is on RNAi and its potential use in the treatment Hep B, the description of RNAi is probably the lowest standard within the manuscript's text.

Many incorrect statements have been made, or a lack of detail has been provided, and together, to me this demonstrates a lack of understanding of small RNA biology and RNAi field by the authors.

Response 1:
Thank you for your valuable feedback on our review. We really appreciate your comments. The manuscript  has been thoroughly revised   and more detailed information has been provided to ensure that it reflects a more accurate understanding of the biology of small RNA  (topic: Discovering an RNAi pathway for gene silencing, fifth, sixth and seventh paragraphs). We have also revised the text to meet the latest scientific consensus on RNAi technology.

Comments 2: There is also numerous other English language issues which require addressing / correction in a revised manuscript version.

Response 2: We conducted a thorough review of the manuscript for linguistic accuracy, improving grammar, phrasing, and overall readability.

Comments 3: 

Figure 3 can be removed as it is inaccurate and fails to provide any additional insight into the general field.

Figure 4 needs work / redoing in order to be mechanistically accurate in the information it is attempting to present.

Please use the annotated copy of your article as a template for correction/amendment as part of producing a revised manuscript version.

Response 3: Figure 3 has been removed as suggested. Figure 4 has been revised to improve its accuracy. However, we would greatly appreciate more specific feedback on what elements in figure 4 require further modification, as we want to ensure it fully meets the standards and is mechanistically accurate. If there are particular details or areas that remain unclear, please indicate them for us.

Round 2

Reviewer 3 Report

Comments and Suggestions for Authors

Manuscript has been improved, but requires additional work via an additional round of review.

Please see annotated PDF for remaining issues to be addressed.

Comments on the Quality of English Language

Another round of careful English editing is required.

Author Response

Comments 1:

Manuscript has been improved, but requires additional work via an additional round of review.

Please see annotated PDF for remaining issues to be addressed.

Response 1: 

Thank you for your feedback and for providing the annotated PDF with detailed suggestions. We have carefully reviewed the document once again and have made additional corrections based on your recommendations. The highlighted issues have been addressed, and the manuscript has been improved accordingly.

Please let us know if any further modifications are needed. We are grateful for your guidance and are open to any additional feedback to ensure the manuscript meets the required standards.

Best regards, 

Pedro Henrique Roque